# Essential Oils as Nematicides in Plant Protection—A Review

**DOI:** 10.3390/plants12061418

**Published:** 2023-03-22

**Authors:** Linda Catani, Barbara Manachini, Eleonora Grassi, Loretta Guidi, Federica Semprucci

**Affiliations:** 1Department of Biomolecular Sciences (DiSB), University of Urbino, 61029 Urbino, Italy; 2Department of Agricultural, Food and Forest Sciences (SAAF), University of Palermo, 90128 Palermo, Italy

**Keywords:** essential oils, nematicidal activity, agriculture, VOSviewer software, Scopus database

## Abstract

By 2030, the European Commission intends to halve chemical pesticide use and its consequent risks. Among pesticides, nematicides are chemical agents used to control parasitic roundworms in agriculture. In recent decades, researchers have been looking for more sustainable alternatives with the same effectiveness but a limited impact on the environment and ecosystems. Essential oils (EOs) are similar bioactive compounds and potential substitutes. Different studies on the use of EOs as nematicides are available in the Scopus database in the scientific literature. These works show a wider exploration of EO effects in vitro than in vivo on different nematode populations. Nevertheless, a review of which EOs have been used on different target nematodes, and how, is still not available. The aim of this paper is to explore the extent of EO testing on nematodes and which of them have nematicidal effects (e.g., mortality, effects on motility, inhibition of egg production). Particularly, the review aims to identify which EOs have been used the most, on which nematodes, and which formulations have been applied. This study provides an overview of the available reports and data to date, downloaded from Scopus, through (a) network maps created by VOSviewer software (version 1.6.8, Nees Jan van Eck and Ludo Waltman, Leiden, The Netherlands) and (b) a systematic analysis of all scientific papers. VOSviewer created maps with keywords derived from co-occurrence analysis to understand the main keywords used and the countries and journals which have published most on the topic, while the systematic analysis investigated all the documents downloaded. The main goal is to offer a comprehensive understanding of the potential use of EOs in agriculture as well as which directions future research should move toward.

## 1. Introduction

Agriculture provides food for billions of human beings; indeed, it is a key sector for the global supply chain and economic development. Considering the rapid population growth, agriculture requires strategies that facilitate increased food productivity, availability, and security [1]. In this sense, the European Union (EU) adopted the European Green Deal that aims to transform agriculture into a more sustainable system through the “Farm to Fork” strategy. In particular, the EU aspires to protect, conserve, and enhance natural capital by mitigating and reducing the impacts of human activities on the environment. This strategy seeks to significantly reduce the use and, consequently, the risks of chemical products by developing and using innovative ways to protect the sustainability of food systems in agriculture [2]. However, finding the best and most sustainable way to solve certain significant problems in agriculture, such as pest control, is challenging. Although invisible to the human eye due to their small dimensions (250 µm to 12 mm in length and 15 to 35 µm in width), nematodes are important factors that can affect and limit crop production [3]. In fact, nematodes are one of the main causes of global annual crop losses that produce million-euro deficits [4]. The present management of nematodes in agriculture, permitted by various national regulatory schemes, uses chemically active ingredients such as carbamate, aldicarb, oxamyl, 1,3 dichloropropene (1,3-D), among others. However, despite their immediate advantages, these chemical products pose potential risks and problematic consequences to the environment due to the application methods used and their diffusion in soil and water, with indirect effects on living organisms’ health and ecosystems, including those of humans [5]. As soon as nematicides are applied to the soil, the degradation (microbial, chemical, or physical) of chemical products begins; some soil microbes play beneficial roles because they allow nematicides to be broken down into environmentally benign molecules, while leaching or surface run-off losses are the most undesirable forms of degradation, as they may enable nematicides to enter groundwater [6]. More often, it is necessary to discover pest control methods that simultaneously have successful effects and a low impact on the ecosystems and their inhabitants. For these reasons, many active chemical compounds have been banned in a number of countries recently, and the chemical products commercially available against phytoparasitic nematodes are scant. Thus, there is an increasing interest in finding resources of natural origin to replace synthetic products. Currently, there is increasing interest in investing in green technologies in many fields, agricultural pest control included. The search for alternative nematode control methods has bet on the use of plants and the natural chemical compounds in them. A great example of this is essential oils (EOs) which, thanks to their properties, have already been deployed against insects, bacteria, fungi, and nematodes. Plant EOs may be a widely available green resource, and their degradation into non-toxic products does not evince any harmful effects on non-target organisms or the environment [7]. EOs are products obtained by mechanical extraction or hydro-distillation from aromatic plants. Plants, in fact, secrete secondary metabolites as a defense strategy that makes them competitive in their own environment [8]. EOs are formed by a variety of volatile compounds which give them their peculiar aroma and chemical compositions; common components include terpenes, sesquiterpenes, phenolic compounds, ketones, acids, and esters [9]. So far, research has tested their efficacy mainly in vitro, with little in vivo testing. EOs have been exploited for their pesticide potential, as there are many reports on their fungicidal [10], anti-microbial [11] and insecticidal activities [12], but nematological research on the use of EOs is a rather new area being developed [13]. Although there are several studies on the use of EOs as potential nematicides, there is a lack of critical evaluation of which EOs should be used as nematicides, and how. Thus, the current review seeks to help those who want to use EOs with nematicidal effects in agriculture by answering the following questions: (1) which is the most studied target nematode, and why? (2) Which are the most used EOs in agriculture and for which nematode, and what is the most preferable formulation in use? From this study, we conclude by proposing future research to better understand the action mechanisms of EOs in nematodes so as to determine how nematicide products may be used in agriculture. Indeed, there are very promising advances to be made in the scientific field in the near future.

## 2. Results and Discussion

### 2.1. Which Is the Most Studied Target Nematode, and Why? A Bibliometric Network Analysis

From the data downloaded in the CSV file, the total number of documents subjected to bibliometric analysis through VOSviewer software was 176. The temporal span of the research extended from 1985 to 2022, with 2021 marked as the most productive year (Figure 1). It is notable that the topic gained increasing attention among the scientific community, especially in the last four years of the period. Among the 36 countries detected, Brazil headed the ranking for the number of articles (26), and this is likely related to the intensified use of its agricultural lands in the last decades, which inevitably leads to emerging nematological problems for different crops [14]. Indeed, over time, Brazil has recorded a succession of diseases caused by different nematode species found in tropical and subtropical regions, together with susceptible crops or varieties (e.g., soybeans, coffee, tomatoes, cotton).

Keywords were grouped into four clusters (Figure 2). The keywords ‘*Origanum vulgare*’ and ‘terpene’ are the only ones that refer to EO origin and composition and are strictly linked together. Thus, the EO of *O. vulgare* is rich in thymol and carvacrol, two natural monoterpenes derived from cymene [15,16]. Generally, the presence of these terpenes makes an EO highly nematicidal, as demonstrated both in vitro [17,18] and in vivo [19].

In the network map, two species of nematodes appeared: ‘*Meloidogyne incognita*’ and ‘*Bursapheluncus xylophilus*’. Answering the first question, it is possible to affirm that these two species are the most studied, based on their scientific and economic importance. Both species are plant-parasitic nematodes (PPNs) which pose the most serious damage and economic losses. In particular, *M. incognita* is responsible for agricultural problems that can affect food security by undermining crops yields, while *B. xylophilus* infects pine trees, causing ecosystem-scale devastation [20].

In agriculture, the genus *Meloidogyne* represents the most widespread and economically harmful pest [3]. These obligate PPNs are also known as root-knot nematodes (RKNs) because they reproduce and feed within plant roots, inducing galls and/or root-knots. Since RKNs claim a worldwide distribution, about 2000 plants, deriving from various habitats, are susceptible to infection by RKNs, which are responsible for approximately 5% of global crop losses [21]. *Meloidogyne* includes more than ninety species [22], but five are considered the most important on a global basis: *M. javanica*, *M. arenaria*, *M. incognita*, *M. hapla*, and *M. graminicola* [23]. Generally, *Meloidogyne* occurs in a wide range of climates, from tropical to subtropical regions, as well as mild temperate zones. For many years, soil fumigation has played a crucial role in controlling RKNs in the production of vegetable crops throughout world. These species have been controlled in fields also by using other chemical products (non-fumigant compounds). However, in recent decades, several chemical products have been withdrawn from the market due to their potential risks to the environment and human health [24]. Other management control practices are used with contrasting results, especially when the density of RKNs is high, including polyethylene mulch, control practices (e.g., rotations to nonhost crops), the use of biocontrol agents (e.g., *Pasturia penetrans*, *Trichoderma* spp.), hot water treatments, biofumigants, and biofumigant crops [6,20,22]. The *Bursaphelenchus* genus is largely known for *B. xylophilus*, the uncontestably most devastating nematode in forestry systems [25], considered a quarantine pest of the EU according to Directive 77/93/EEC. *B. xylophilus* is also known as the pine wood nematode (PWN) for the significant damage it causes to pine forests [26]. Pine wilt disease (PWD) is a serious threat to forests and consequently to ecosystems. PWN is considered one of the most important pests in the world, and many control methods have been applied [27]. Although burning infected trees is the most efficient method to control PWN, it is also the most unsustainable.

### 2.2. Which Are the Most Used EOs in Agriculture and for Which Nematodes, and What Is the Most Preferable Formulation Used? A Systematic Analysis of the Literature

After the selection described in Figure 3, a total of 63 documents were put under scrutiny for the systematic review; all are reported in Appendix A. As reported by the VOSviewer keyword results, *Meloidogyne* and *Bursaphelenchus* were the most investigated genera.

The data and main information on the effects of the various EOs examined against the different nematode target species are summarized in Table 1. The results identify the concentrations at which the EOs, or possibly the liquid blends in which they were mixed, evidenced signs of nematicidal effects on the nematode population. Mortality, paralysis, and or hatching inhibition of nematodes were considered as relevant nematicidal effects.

Among the different EOs tested on *B. xilophilus*, EOs derived from plants belonging to the Lamiaceae family were the most investigated (*Thymus vulgaris*, *Satureja montana*, and *Thymbra capitata*), whereas Triton X-100 was shown to be the best solvent in EO bioassays. In fact, it is known for its capacity to dilute oils, ability to increase tissue permeability, and relative ease of handling. However, its uses have some drawbacks that should be taken into consideration; examinations showed that some EOs are difficult to dissolve in Triton X-100, and its use is often subjected to higher variability effects compared to other solvents. Acetone was investigated and found to be an adequate alternative to Triton X-100, especially better suited for EO dilution [81]. Overall, it is possible to state that the most effective nematicidal effects on *B. xylophilus* were obtained using *Allium sativum* (Liliaceae). In particular, 100% mortality was recorded after 4 h of exposure to a solution of distilled water (DW) containing Triton X-100 (5000 ppm) and an EO concentration of 62.5 μL/L. From gas chromatography–mass spectrometry (GC-MS) analysis it emerged that diallyl trisulphide, followed by diallyl disulphide, cinnamyl acetate, and cinnamaldehyde, were the most toxic components in the EO extracted from garlic bulbs. *Ruta graveolens* (Rutaceae), *S. montana*, and *T. capitata* were, however, active EOs after 24h of exposure because lethal concentration (LC100_24_) was achieved at concentrations <0.4 µL/mL. Here, EOs were prepared in methanol solution at 40 μL/mL; methanol was substituted for the conventional solvent Triton in X-100 because of its higher polarity and capacity. Specifically, *S. montana* and *T. capitata* EOs presented high levels of carvacrol, ϒ-terpinene, and p-cymene, judged so far as being responsible for the nematicidal activity [35].

The strongest nematicidal activity on *M. incognita* juvenile stage 2 (J2) was achieved with cinnamon and garlic EOs. Both EOs were studied by the same research group headed by Jardim (2018 with *Cinnamon cassia* (Laureaceae) EO, and 2020 with *A. sativum*) who set up the same experiments under in vitro conditions. To obtain aqueous solutions of EO, the oil extracted from the bark and bulb, respectively, was emulsified in a water solution with 0.01 g/mL Tween 80 to result in a final concentration of 10,000 µg/mL. This primary emulsion was then diluted to different concentrations. With *C. cassia* EO, concentrations of 250, 125, and 62 µg/mL were lethal to *M. incognita* J2, while with garlic EO, the nematode population was slightly more resistant as mortality was recorded with concentrations of EO of 500, 250, and 125 µg/mL. *C. cassia* is mainly composed of (E)-cinnamaldehyde that is primarily responsible for the activity against the nematode, followed by o-methoxycinnamaldehyde and benzaldehyde. This aldehyde showed efficacy similar to carbofuran, a commercial nematicide used to reduce several nematode populations [59]. Instead, the nematicidal activity of garlic EO is attributed, as reported above, to organosulfur compounds [63].

A mixture (1:1) of *Haplophyllum tuberculatum* (Rutaceae) and *Plectranthus cylindraceus* (Laminaceae) EOs was the best treatment against *M. javanica*. The combination of the two EOs was highly toxic after 24 h of exposure time, at 12.5, 25, and 50 µg/mL. The solutions were obtained by diluting 20 mg of each oil with 100 mL of 0.01% Tween 20, then mixing 50 mL of each oil together, obtaining a mixture solution at 100 µg/mL. The mixture of *H. tuberculatum* and *P. cylindraceus* EOs was comparable to the toxicity of carbofuran because the synergic presence of the alkene limonene and the phenol carvacrol produced a better nematicidal tool to control RKN at lower concentrations [67]. Strong mortality effects on *M. javanica* J2 were also obtained with 1 mg/mL of *Thymus satureioides* (Lamiaceae), *M. spicata*, and *Lippia citridora* (Myrtaceae) EOs after 72 h of exposure. *M. spicata* showed the strongest effects on J2 mortality already at 24 and 48 h. The main constituents of these active oils are monoterpenes (carvone, limonene, menthol, thymol) which, taken alone, do not exhibit nematicidal activity, but the synergic interaction of carvone with either limonene or menthol seems to have significant effects on *M. spicata*. Instead, the activity of *T. satureioides* EO is referred to the presence of thymol [17], and the exposure solutions tested (ranged from 1 to 0.5 mg/mL) were obtained by dissolving the EOs in DW containing 5% of a DMSO-Tween solution (0.5% Tween 20 in DMSO) [70].

The PPNs of the genera *Criconemella* spp., *Hoplolaimus* spp. and of the species *Rotylenchulus reniformis* showed sensibility to *T. vulgaris* and *Mentha spicata* (Lamiaceae) EOs because the nematode population’s motility stopped (vitality absence) after 72 h. Particularly, the EO of each plant was diluted at 0.05 and 0.10 with DW using 0.05% Tween 80 as a spreading agent. Gas-liquid chromatography (GLC) analysis demonstrated that *M. spicata* is mainly composed of carvone (58%), while *T. vulgaris* is mostly p-cymene (41%) and thymol (19%). In this study, the results supported the idea that the presence of different chemical compounds may lead to different and unknown action mechanisms on nematodes [37].

The in vitro nematicidal effects of four aromatic plants were evaluated by Avato et al. (2017) on two other important PPNs, such as the migratory endoparasite *Pratylenchus vulnus* which has over 80 hosts, including fruits, nuts, peaches, grapevines, soybeans, and many woody perennials, and the dagger nematode *Xiphinema index* that has high economic impact in vineyards by direct pathogenicity and, above all, by transmitting the grapevine fanleaf virus (GFLV). Both species of nematodes were exposed for 24, 48, and 96 h to different EO concentrations (2, 5, 10,15, 30 µg/mL) of *Artemisia herba-alba* (Asteraceae), *Citrus sinensis* (Rutaceae), *Rosmarinum officinalis* (Lamiaceae), and *T. satureioides*, obtained by adding a 0.3% water solution of Tween 20. *X. index* was highly sensible to *A. herba-alba*, *R. officinalis*, and *T. satureioides* EOs because, after only 24 h, the lowest concentration (2 µg/mL) was enough to completely kill the nematode population; the situation persisted at all times and concentrations tested. Instead, only 75% mortality was reached on *P. vulnus* by *R. officinalis* at 15 µg/mL concentration, and at 96h *P. vulnus* evidenced a stronger resistance compared to other PPNs. The different stress caused by the various EOs on the PPNs suggested the possible involvement of different reaction mechanisms associated with the anatomy and feeding behavior of the nematodes. All the EOs analyzed by Avato et al. (2017) are made up of monoterpene constituents; thujone and camphor caused nematicidal activity in *A. herba-alba*, 1,8- cineole, a-pinene and camphor in *R. officinalis*, and borneol and thymol in *T. satureioides* [56].

In all formulations selected as having nematicidal effects, the emulsifiers Tween 80 or 20 were used. Both are known as polysorbates, popular non-ionic detergents commercially available under the name Tween and employed for their useful features such as great accuracy, ease of use, purity, and stability. Although they can be often used interchangeably, they differ in their chemical formulas and possible applications [82].

*P. anisum* and *O. vulgare* EOs were totally lethal to the J2 of the false root-knot nematode *Nacobbus aberrans* (formally a species complex, with many pathotypes having different host preference) after 24 h. The obtained oils, from dried oregano leaves and dried anise seeds, respectively, were considered as standards at 100%, then diluted using 2% DMSO in water for testing suitable concentrations. DMSO (dimethyl sulfoxide) is a broadly commercialized and used solvent. The concentrations used for testing the *N. aberrans* response varied from 10 up to 5000 µL/L for anise, and from 200 to 5000 µL/L for oregano oil. The highest doses tested (200–5000 µL/L) gave a rate of mortality of 100% after just 2 h of exposure. *O. vulgare* showed the highest toxic effects (mainly due to the high content of carvacrol, i.e., 40%; thymol, 28.1%; and σ-cymene, 13.6%) when doses and time exposure increased. However, the most suitable dose to kill 100% of *N. aberrans* larvae was 600 µL/L. Sosa et al. (2020) underlined that anise (mainly represented by anethole, 89.5%) has a higher bioactivity than oregano, reaching effectiveness at 200 µL/L [68].

Table 2 summarizes all the genera of plants from which the EOs were extracted and used for nematicidal purposes and reports all the EOs that caused 100% mortality of nematode targets, thus having the highest and strongest efficacy among the studies analyzed for the review. It is possible to note that EOs derived from *A. sativum* and *T. vulgaris* can be marked as the most used oils against different PPNs. Several studies have shown that garlic extracts are known to have antibacterial [83], antifungal [84], and nematicidal activities [85]. The abundance of organosulfur compounds such as alliin, allicin (S-allylcysteine sulfoxide) and some polysulfanes (diallysulfide, diallyldisulfide, diallyltrisulfide, diallyltetrasulfide) give garlic EO robust activity against multiple problems and they have been reported as green pesticides [86]. Instead, the combined presence of thymol and carvacrol, two phenolic monoterpenes which provide bioactivity, allows thyme to be the most used and important medical plant in several industries (cosmetics, pharmaceuticals, and food products) [87]. An important role is played by the EOs of *M. spicata* and *M. hortensis* as well. The first, commonly called spearmint, is widely cultivated and its oil is commercialized in many fields. *M. spicata* has several biological uses thanks to the abundance of carvone, a potent monoterpene able to suppress fungal disease and the sprouting of stocked foods [88]. *M. hortensis* contains many secondary metabolites, terpenes predominantly, with valuable biological activity; considerable studies have demonstrated its pesticide activity [89]. Finally, it is possible to affirm that EOs derived from plants belonging to the Lamiaceae family are the most investigated. In addition to the varied bioactivities of EO compounds in Lamiaceae, there is also the importance of its widespread distribution and easy versatility of use [90].

## 3. Materials and Methods

This review study derived from a union of bibliometric and systematic analyses performed by collecting documents derived from the Scopus database [91], the largest database of scientific peer-reviewed literature. The research on Scopus was conducted through the association of two keyword terms: “essential oil” and “nematode,” which are at the core of our research focus. The Boolean operator AND, put between the keywords, told the database that both search terms needed to be present in the resulting articles. The documents thus obtained were included in the period that began from the evidence of the first document available on the Scopus database to 1 September 2022. The bibliometric analysis aims to visualize network maps that, using VOSviewer software, summarized large volumes of scientific data. The systematic review, instead, summarized data from primary research tools, in this case, scientific articles on the agricultural use of EOs against nematodes. The literature research was conducted by following a selection process exhaustively described in Figure 3. The obtained articles were screened in two-steps to ensure their relevance.

In the first step, all the documents obtained from Scopus were selected for the bibliometric analysis. The research returned 355 articles, and the CSV file was analyzed using the VOSviewer software.

In the second step, the full texts of all articles were critically read, and those not relevant to the main research goal of the review (e.g., articles that deal with EOs against house flies, *Anisakis simplex*, food preservation, bacteria) were then discarded, including records marked as reviews, records which investigated terpene functions or other chemical compounds rather than EOs, records where text was unavailable, or records that were duplicates or had not been published in English.

### Bibliometric Network Analysis

VOSviewer software (version 1.6.8, Nees Jan van Eck and Ludo Waltman, Leiden, The Netherlands) was the tool used to perform all the bibliometric network analyses [92]. VOSviewer software works with a CSV file, which can be downloaded from Scopus. VOSviewer can generate different types of maps and networks based on the visualization of clusters composed of nodes or items linked by connections. The main technical terms, together with the different types of analysis performed, are summarized in Table 3. It is important to specify that the size of nodes is determined by different weight attributes, total link strength, number of documents, and number of citations. In this study, analyses of (1) both cluster and overlay visualization for research on co-authorship among countries, (2) keyword co-occurrence, and (3) cited scientific journals were performed. The overlay visualization differs from the cluster (association of keywords) in that the maps are weighted based on the average number of citations and average year of publication. A thesaurus file was created especially for the co-occurrence of keywords in order to avoid repetitions and synonyms among the keywords and allow the unification of terms (e.g., essential oil/essential oils were unified in the singular form essential oil), making the network map more readable.

## 4. Conclusions and Future Prospects

European Legislation (Reg. CE 396/2005 and 1095/2007) has recently been revised, restricting the use of pesticides on agricultural crops, mainly because of increasing awareness of potential risks to the environment as well as to human and animal health. In response, several green alternatives, such as EOs and plant extracts, have been catching on for the control and protection of plants and crops against attacks of PPNs in agriculture.

This review describes how EO formulations have been used to date on nematodes, and in particular, which EOs have shown greater nematicidal activity. From the literature available in the Scopus database, most research has focused on the genera *Meloidogyne* and *Bursaphelencus*, known for causing significant damage to economically important crops worldwide. The PPNs studied showed significantly less resistance to EOs from garlic (*A. sativum*) and thyme (*T. vulgaris*), marking them as EOs with the highest nematicidal activity. This activity is supported by recent studies which have indicated that garlic-based nematicides could be an effective tool for *X. index* management in organic and integrated vineyards [93].

The careful identification and isolation of EO chemical components is necessary in order to determine the possible synergic effects of the “mixed” components and understand their action mechanisms so as to exploit them in commercial fields. However, the terpenes, organosulfur, and phenolic compounds reported in Table 4 seem to be the main components responsible for nematicidal activity on the various target species.

Despite the attention EOs have drawn from the scientific community and numerous in vitro studies which have underlined a wide spectrum of their applications as nematicides, so far, few attempts have been made to use them in plant protection; this lack is demonstrated by the surprisingly scarce number of homologated EO formulas. The high volatility of EOs and their high costs are likely among the reasons for this, making it necessary to invest more in new and inexpensive formulations and processes that will permit their successful application in the field. A win-win strategy that aims to shorten the distance between scientific research and politics is likely crucial. Thus, technical consultation among different researchers (i.e., agronomists, nematologists, biochemists, and economists), farmers, and EO producers is recommended to overcome these current gaps.

In conclusion, future research should investigate EO formulations on a wider scale to enhance their potentialities, modes of action, cost-effectiveness, and potential impacts on non-target organisms.

## Figures and Tables

**Figure 1 plants-12-01418-f001:**
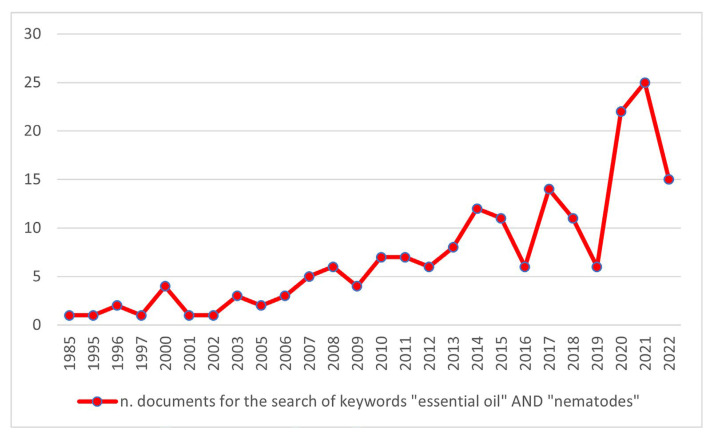
The total number of documents for the search of keywords “essential oil” AND “nematodes” in the Scopus database.

**Figure 2 plants-12-01418-f002:**
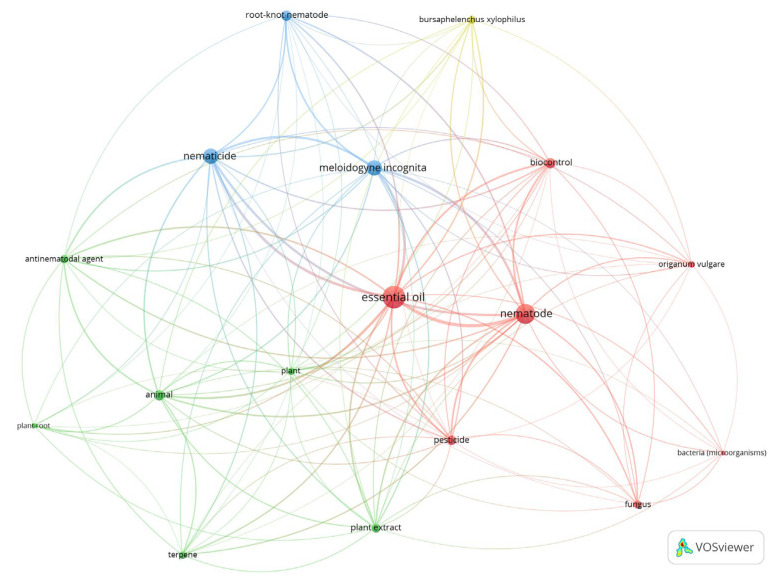
Network map of keyword co-occurrences for “essential oil” AND “nematodes”. The number of keyword co-occurrences was 1322; a thesaurus file was necessary to reduce them to 1257. A total of 17 keywords met the threshold and were divided into 4 clusters (plot created by VOSviewer software).

**Figure 3 plants-12-01418-f003:**
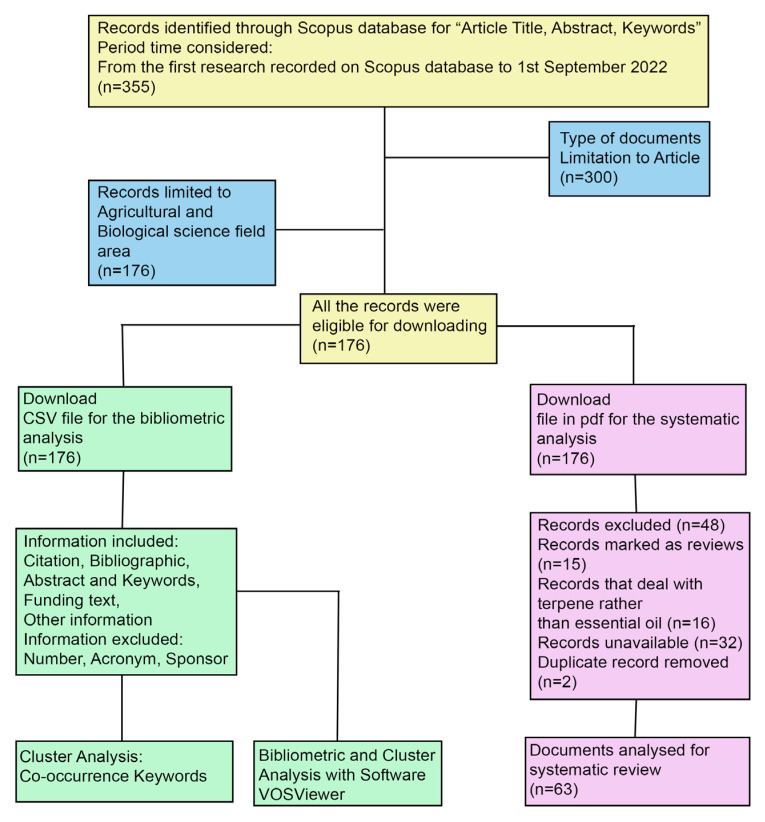
Flow diagram for article selection. The process highlights the number of studies identified along with exclusion and inclusion criteria.

**Table 1 plants-12-01418-t001:** Most significant results of the literature analysis. Each publication shows the essential oil(s) (EO) and the nematode target. Each line of the table corresponds to a single article in which significant nematicidal effects (mortality, paralysis, hatching inhibition) are reported together with the EO concentration and time exposure. The multiple “*, **, ***, ****” are used to diversify the results in each article, therefore in each table’s line, where different EOs showed results at varied concentrations. It is possible that the article is repeated as different nematodes (genus or species) might be put under scrutiny.

Nematode Targets	Botanical Species of Essential Oil (EO)	Effect on Nematodes	Time Exposure	Type of Assay	EO Concentration and Type of Solvent	References
*Bursaphelenchus* *B. xylophilus*	*Allium sativum* *, *Cinnamomum verum* **	100% mortality	4 h	in vitro	62.5 * μL/L and 125 ** μL/L EO + Triton X- 100 (5000 ppm)	[28]
	*Thymus vulgaris* (red * and white **)	LC50 (mg/mL)	24 h	in vitro	1.39 mg/mL *, 1.64 mg/mL ** EO + 1 μL of polyoxyethylene hydrogenated castor oil-ethanol solution (1 mg/mL)	[29]
	*Litsea cubeba* *, *Trachyspermum ammi* **, *Pimienta doica* ***	LC50 (mg/mL)	24 h	in vitro	0.504 *, 0.431 **, 0.609 *** mg/mL EO + DW containing Triton X-100 (5000 ppm)	[30]
	*Syzygium aromaticum*, *T. vulgaris*	100% mortality	24 h	in vitro	EO (0.1 mL/L) + DW containing Triton X-100 (5000 ppm)	[31]
	*Coriandrum sativum*, *Liquidambar orientalis*, *Valeriana wallichii*	100% mortality	24 h	in vitro	EO 2.0 mg/mL + Triton X-100 (5 mg/mL)	[32]
	*Cinnamomum zeylanicum*	LC50 (mg/mL)	24 h	in vitro	0.12 mg/mL EO (1 mg EO dissolved in 1 µI of ethanol-Triton X-100 solution (9:1 by volume)	[33]
	*Chamaespartium tridentatum*, *Origanum vulgare*, *Satureja montana*, *Thymbra capitata*, *Thymus caespititius*	100% mortality	24 h	in vitro	2 mg/mL solution + Triton X-100 in DW solution (5 g/mL)	[34]
	*Ruta graveolens*, *S. montana*, *T. capitata*	100% mortality	24 h	in vitro	<0.4 µL/mL EO + methanol 40 µL/mL	[35]
*Ditylenchus* *D. dipsaci*	*Eugenia caryophyllata* **, *Origanum compactum* *, *O. vulgare* *, *Thymus matschiana* *, *T. vulgaris* *	>95% * mortality, >80% ** mortality	3 h	in vitro	5000 and 7500 ppm EO + 10% ethanol (*v*/*v*) diluted in water containing 0.3% Tween 20 (*v*/*v*)	[36]
*Criconemella* spp.	*Majorana hortensis **, *Mentha longifolia ** Mentha spicata **, *T. vulgaris **	100% * and 88% ** reduced motility	72 h	in vitro	0.05 and 0.10 EO (0.05% Tween 80 and water)	[37]
*Meloidogyne* spp.	*Melaleuca alternifolia*	100% mortality	20, 20, and 16 h, respectively to concentrations	in vitro	5, 10, 15 mg/mL EO + Tween 20 (1%),	[38]
	*Thymus citriodorus*	EC50 0.09 * *v*/*v*, 0.08 ** *v*/*v* paralysis	24 *, 48 ** hours	in vitro	dried *T. citriodorus* with DW at a ratio of 1/10 (*w*/*v*)	[39]
*Meloidogyne* *M. arenaria*	*M. piperita*, *M. spicata*	reduced >50% the number of galls	1 week	in vivo	1500 mg oil/kg soil	[40]
	*Petroselinum crispum*	EC50 (mg/L) J2 paralysis	72 h	in vitro	416 mg/L EO	[41]
*Meloidogyne* *M. artiellia*	*Chrysanthemum coronarium*	65% mortality	10 days	in vivo	4, 8 and 16 μL/mL	[42]
*Meloidogyne* *M. chitwoodi*	*Cymbopongon citraturs*, *Foeniculum vulgare*, *T. caespititius*, *Thymus zygis*,	≥90 % hatching inhibition	24 h	in vivo	2 µL of EO in methanol	[43]
*Meloidogyne* *M. graminicola*	*C. citratus Ocimum*, *basilicum*, *M. piperita*	>80% mortality of second stage juveniles	24 h	in vitro	1% EO: DW containing ethanol 0.5% (*v*/*v*) and Triton X-100 0.5% (*v*/*v*)	[44]
*Meloidogyne* *M. hapla*	*P. crispum*	EC50 (mg/L) J2 paralysis	72 h	in vitro	611 (mg/L) EO	[41]
*Meloidogyne* *M. incognita*	*Eucalyptus citriodora*, *Eucalyptus hybrida*, *O. basilicum*	100% mortality larvae	24 h	in vitro	250, 500, 1000 ppm EO + 0.5 mL (DMSO) with 0.5% Tween80 and DWr	[45]
	*F. vulgare **, *Eucalyptus* spp ****, *Origanum syriacum ***, *Pinus pinea ***	>85% *, >60% ** mortality J2	24 h	in vitro	100 mg/L EO in 2 mL of DW	[17]
	*Pilocarpus microphyllus*	95% mortality	24 h	in vitro	1 mg EO and water: DMSO (98:2) solution to complete 1 mL (1000 ppm as final concentration)	[46]
	*S. aromaticum*	EC50 0.097% EO (*v*/*v*) egg hatching inhibition, EC50 = 0.104% EO (*v*/*v*) J2 viability	48 h	in vitro	1 mL EO in an aqueous carrier solution of 0.25% L-α-phosphatidylcholine (lecithin) from soybean (Sigma-Aldrich, Munich, Germany) + Triton X-114 (0.1%)	[47]
	*A. sativum **, *Azadirachta indica ***, *Eucalyptus chamadulonsis **, *Tagetes erecta ***,	40% *, >60% ** mortality Juveniles	24 h	in vitro	0.05% EO diluted with tap water	[48]
	*A. sativum*, *T. vulgaris*	reduced number of egg mass on plants and root galling formations	65 days	in vivo	50 µL/plant	[49]
	*Eucalyptus meliodora* *, *F. vulgare* **, *Pimpinella anisum* **, *Pistacia terebinthus* ****	EC50 (μg/mL) J2 paralysis	96 h	in vitro	807 *, 231 **, 269 ***, 1116 **** μg/mL + ethanol and Tween20 diluted 1 and 0.3% (*v*/*v*), respectively	[50]
	*Ruta chalepensis*	EC50 (mg/L) J2 paralysis	24 h	in vitro	77.5 mg/L EO + ethanol 1% (*v*/*v*) and Tween 20 0.3% (*v*/*v*)	[51]
	*M. piperita **, *Mentha pulegium ***, *M spicata ****	EC50 (mg/L) J2 paralysis	72 h	in vitro	1005 *, 745 **, 300 *** mg/L EO + methanol and Tween 20 in each well never exceeded 1 and 0.3% (*v*/*v*), respectively	[52]
	*Agastache rugosa*	LC50 (μg/mL)	72 h	in vitro	47.3 μg/mL EO + diluted in water and 2% DMSO	[53]
	*P. crispum*	EC50 (mg/L) J2 paralysis	72 h	in vitro	140 mg/L EO	[41]
	*E. citriodora ***, *E. globulus **, *M. piperita ***, *Pelargonium asperum **, *R. graveolens ***	sensible reduction of gall formation and number of nematode eggs J2 compared to non-treated soil rates	60 days	in vivo	50 *, 200 ** μL/kg soil rates of pure EO (soil fumigation)	[54]
	*Artemisia herba-alba **, *R officinalis ***	94% *, 98% ** mortality	24, 96 h	in vivo/in vitro	15 µg/mL EO (0.3% water solution of Tween 20)	[55]
	*A. annua*	100% mortality of J2	24 h	in vitro	500 and 250 ppm EO	[56]
	*Monarda didyma*, *Monarda fistulosa*	LC50 (μL/mL) of juveniles	24 h	in vitro	1.0 μL/mL EOs + 0.3% Tween 20 in water solution	[57]
	*Cinnamon cassia*	100% mortality of J2	48 h	in vitro	62 µg/mL EO + 0.01 g/mL Tween 80 in water	[58]
	*C. zeylanicum*	LC50 *, LC95 ** (μg/mL)	48 h	in vitro	49 * and 131 μg/mL ** + 0.01g/mL Tween 80 (concentration 400 μg/mL)	[59]
	*Cymbopogon schoenanthus **, *C. zeylanicum ***, *Ocimum sanctum ****	LC50 (mg/L)	24 h **, 48 h (*) (***)	in vitro	288 *, 391 **, 282 *** mg/L EO + DW with Tween 20 (never exceed 1 and 0.3%, respectively)	[60]
	*Artemisia nilagirica*	LC50 (µg/mL)	48 h	in vitro	5.75 μg/mL + 0.3% Tween 20	[61]
	*C. verum **, *E. citriodora ***, *R. graveolens ****, *Syzygium aromaticum *****	LC50 (µg/mL)	24 h	in vitro	0.1 *, 1.6 **, 1.4 ***, 1.8 **** µg/mL EO + 0.3% Tween20	[62]
	*A. sativum*	100% immobility and mortality of J2	48 h	in vitro	125, 250, 500 μg m/L EO + 0.01 g m/L Tween 80 (final concentration of 10000 μg m/L)	[63]
	*Satureja hellenica*	100% paralysis * and 100% mortality of J2 **	96 *, 48 h **	in vitro	2000 µL/L *, 4000 µL/L **	[64]
	*Brassica nigra*	LC50 (μg m/l)	72 h	in vitro	50, 75, 100 μg m/L EO + DW + Triton X-100 (2%, *w*/*v*)	[65]
	*Acorus calamus *. C. sinensis ***, *M. alternifolia ****	LC50 (µg/mL)	72 h	in vitro	85.23 *, 39.37 **, 76.28 ***(µg/mL) EO + Atlas G5002 surfactant (2% *w*/*w*)	[66]
*Meloidogyne* *M. javanica*	*Carum carvi*, *F. vulgare*, *Mentha rotundifolia*, *M. spicata*	Inhibited hatching and immobilized nematodes	7 days	in vivo	800 and 600 μL/L EOs (10% ethanol, *v*/*v*) were diluted with water containing 0.3% (Tween 20 *v*/*v*)	[16]
	*Haplophyllum tuberculatum*, *Plectranthus cylindraceus*	100% mortality	24 h	in vivo/in vitro	a mixture of the two EOs (1:1) at 12.5, 25, 50 µg/mL (0.01% Tween 20)	[67]
	*R. chalepensis*	EC50 (mg/L) paralysis of J2	24 h	in vitro	107.3 mg/L EO + ethanol 1 (*v*/*v*) and Tween 20 0.3% (*v*/*v*)	[51]
	*Eupatorium viscidum*	paralysis of J2	72 h	in vitro	1 µg/mL EO	[68]
	*R. officinalis*	no effect on the population	1 year in soil	in vivo/in vitro	0, 1, 2, 3% + 1% plant oil additive (Natur’l Óleo, Stoller, Sao Paulo, Brazil)	[69]
	*Lippia citrodora*, *M. spicata*, *Thymus satureioides*	100% mortality	72 h (M. spicata 24, 48 h too)	in vitro	1 µg/mL EO + 5% of a DMSO-Tween solution (0.5% Tween 20 in DMSO)	[70]
	*F. vulgare*	LC50 (µg/mL)	48 h	in vitro	500, 1000, 2000, 3000 µg/mL EO	[71]
	*Piper hispidinervum*	100% mortality of J2	72 h	in vitro	EO 1 mg/mL dissolved in DW containing 5% of a DMSO-Tween solution (0.5% Tween 20 in DMSO)	[72]
	*Artemisia absinthium*	100% mortality of J2	5 days	in vivo/in vitro	100 and 50% of *A. absinthium* hydrolates (extracted with activated carbon) and their organic fraction was dissolved in water with 5% of DMSO-Tween 20 solution (0.5% Tween20 in DMSO) at 20 mg/mL	[73]
	*Tagetes minuta*	100% mortality of J2	24, 48, 72 h	in vitro	5 µL EO diluted in a DMSO-Tween 20 solution (0.5% Tween 20 in DMSO)	[74]
	*Schinus terebinthifolius* (green fruits)	reduced hatching by 86% and increased juvenile mortality by 300%	24 h	in vitro	100 μL EO + 100 μL of 0.3% Tween 20 aqueous solution	[75]
	*S. montana*	100% mortality of J2	72 h	in vitro	0.12 μg/μL EO dissolved in DW containing 5% of a DMSO Tween solution (0.6% Tween 20 in DMSO)	[76]
	*S. hellenica*	100% paralysis * and 100% mortality of J2 **	96 *, 48 h **	in vitro	2000 µL/L *, 4000 µL/L **	[64]
	*Ridolfia segetum*	71% immobility of J2 and <10% mortality	72 h	in vitro	16 μL/mL EO + water + 0.1% Tween 20 (*v*/*v*)	[77]
	*A. sativum*	100% mortality of J2	24, 48, 72 h	in vitro	Hydrolates + EO at 20 mg/mL were dissolved in a 5% DMSO-Tween solution in water (0.5% Tween 20 in DMSO)	[78]
*Hoplolaimus* spp.	*M. hortensis*, *M. spicata*, *T. vulgaris*	100% mortality	72 h	in vitro	0.05 and 0.10 EO (0.05% Tween 80 and water)	[37]
*Rotylenchulus* *R. reniformis*	*M. hortensis*, *M. spicata*, *T. vulgaris*	100% mortality	72 h	in vitro	0.05 and 0.10 EO (0.05% Tween 80 and water)	[37]
	*A. annua*	100% mortality	24 h	in vitro	500 and 250 ppm EO	[57]
*Panagrolaimus* spp.	*C. burmannii **, *C. cassia ***	LC50 µL/mL	24 h	in vitro	0.033 *, 0.034 ** µL/mL EO	[79]
*Nacobbus* *N. aberrans*	*O. vulgare* *, *P. anisum* **	LD100 (μL/L) of juvaniles	24 h	in vitro	600 μL/L *, 200 μL/L ** pure oil with 2% DMSO in sterile DW	[68]
*Pratylenchus* spp.	*E. globulus*	Reduced nematode population	from 4 to 14 weeks after sowing corn	in vivo	10, 20 and 30 mg/kg soil in the field	[79]
*Pratylenchus* *P. brachyurus*	*R. officinalis*	no effect on population	1 year in soil	in vivo/in vitro	0, 1, 2, 3% + 1% plant oil additive (Natur’l Óleo, Stoller, Sao Paulo, Brazil)	[69]
*Pratylenchus* *P. vulnus*	*R. officinalis*	75% mortality	96 h	in vivo/in vitro	15 µg/mL EO (0.3 % water solution of Tween 20)	[56]
	*M. didyma*, *Monarda fistulosa*	LC50 µL/mL	24 h	in vitro	15.7 *, 12.5 ** μL/mL + 0.3% Tween 20 in water solution	[58]
*Caenorhabditis* *C. elegans*	*Amomum subulatum*	LC50 (μg/mL)	24 h	in vitro	341 μg/mL + 50 μL sterile water + DMSO (1%)	[80]
	*Artemisia nilagirica*	LC50 (μg/mL)	48 h	in vitro	8.32 μg/mL + 0.3% Tween 20	[61]
*Xiphinema* *X. index*	*A. herba-alba*, *R. officinalis*, *T. satureioides*	100 % mortality	24, 48, 96 h	in vivo/in vitro	2, 5, 10 and 15 µg/mL EO (0.3 % water solution of Tween 20)	[56]

Notes: h: hours; EO: essential oil; LC50: lethal dose which causes the death of 50% of a group of test animals. EC50: effect concentration of a drug that is necessary to cause half of the maximum possible effect. ppm: parts per million; Triton X-100: non-ionic surfactant mixtures varying in the number of repeating ethoxy (oxy-1,2-ethanediyl) groups. Triton™-100 is a solvent produced and registered by Sigma-Aldrich, Munich, Germany. DW: distilled water; DMSO: dimethyl sulfoxide is an organosulfur compound used as a solvent.

**Table 2 plants-12-01418-t002:** Names of plant species from which essential oils (EOs) were extracted and used on nematodes. The plant family is reported for each EO. The sign “X” indicates a combination between the EO and specific nematode target. The data correspond to the EOs that induced the total mortality (100%) of nematode targets.

Name of Plant Species	Family	*B. xyl*	*C*. spp.	*M. inc*	*M. jav*	*H.* spp.	*R. ren*	*N. abe*	*X. ind*
*Artemisia absinthium*	Asteraceae	-	-	-	X	-	-	-	-
*Artemisia annua*	Asteraceae	-	-	X	-	-	-	-	-
*Artemisia herba-alba*	Asteraceae	-	-	-	-	-	-	-	X
*Allium sativum*	Liliaceae	X	-	X	X	-	-	-	-
*Cinnamon cassia*	Laureaceae	-	-	-	-	-	-	-	-
*Cinnamon verum*	Lauraceae	X	-	-	-	-	-	-	-
*Cymbopogon citratus*	Poaceae	-	-	-	-	-	-	-	-
*Coriandrum sativum*	Apiaceae	X	-	-	-	-	-	-	-
*Chamaespartium tridentatum*	Fabaceae	X	-	-	-	-	-	-	-
*Eugenia caryophyllata*	Myrtaceae	-	-	-	-	-	-	-	-
*Eucalyptus citridora*	Myrtaceae	-	-	X	-	-	-	-	-
*Eucalyptus hybrida*	Myrtaceae	-	-	X	-	-	-	-	-
*Haplophyllum tuberculatum*	Rutaceae	-	-	-	X	-	-	-	-
*Lippia citriodora*	Verbenaceae	-	-	-	X	-	-	-	X
*Liquidambar orientalis*	Altingiaceae	X	-	-	-	-	-	-	-
*Majorana hortensis*	Lamiaceae	-	X	-	-	X	X	-	-
*Mentha piperita*	Lamiaceae	-	-	-	-	-	-	-	-
*Mentha spicata*	Lamiaceae	-	X	-	X	X	X	-	-
*Ocimum basilicum*	Lamiaceae	-	-	X	-	-	-	-	-
*Ocimum vulgare*	Lamiaceae	X	-	-	-	-	-	X	-
*Pimpinella anisum*	Apiaceae	-	-	-	-	-	-	X	-
*Plectrhantus cylindraceus*	Lamiaceae	-	-	-	X	-	-	-	-
*Piper hispidineryum*	Piperaceae	-	-	-	X	-	-	-	-
*Ruta graveolens*	Rutaceae	X	-	-	-	-	-	-	-
*Rosmarinum officinalis*	Lamiaceae	-	-	-	-	-	-	-	X
*Syzygium aromaticum*	Myrtaceae	X	-	-	-	-	-	-	-
*Satureja hellenica*	Lamiaceae	-	-	X	X	-	-	-	-
*Satureja montana*	Lamiaceae	X	-	-	X	-	-	-	-
*Tagetes minuta*	Asteraceae	-	-	-	X	-	-	-	-
*Thymus caespititius*	Lamiaceae	X	-	-	-	-	-	-	-
*Thymus satureioides*	Lamiaceae	-	-	-	-	-	-	-	-
*Thymus vulgaris*	Lamiaceae	X	X	-	-	-	X	-	-
*Tymbra capitata*	Lamiaceae	X	-	-	-	-	-	-	-
*Valeriana wallichii*	Valerianaceae	X	-	-	-	-	-	-	-

Notes: *B. xyl.*: *Bursaphelenchus xylophilus*; *C*. spp.: *Criconemella* spp; *M*. inc.: *Meloidogyne incognita*; *M. jav*.: *Meloidogyne javanica*; *H*. spp: *Hoplolaimus* spp.; *R. ren*.: *Rotylenchulus reniformis*; *N. abe*.: *Nacobbus aberrans*; *X. ind*.: *Xiphinema index*.

**Table 3 plants-12-01418-t003:** Terminology and different types of analysis performed by VOSviewer software [92].

Terms	Description
Items	Objects of interest (i.e., publications, keywords, researchers, journals)
Link	Connection or relation between two items (i.e., co-occurrence of keywords)
Network	Set of items connected by their links
Cluster	Sets of items included in a map. One item can belong only to one cluster
Link strength	Value of each link, expressed by a positive or negative numerical value. In the case of co-occurrence keyword links, the higher the value, the higher the number of occurrences for that keyword
Number of links	The number of links of an item with other items
Total link strength	The additive strength of the links of an item with another item
Co-occurrence analysis	The analysis of co-occurrence of two keywords linked together with the number of publications in which both keywords occur simultaneously in the title, abstract or keyword list

**Table 4 plants-12-01418-t004:** Names of plant species from which essential oils (EOs) were extracted and used on nematodes. Percentages of main chemical components (considered with a percentage >10%) derived from chemical analysis, and related chemical structure configuration, when required by the study.

Plant Name and Essential Oil	Nematode Target	Main Chemical Components and Percentages	References
*Allium sativum*	*B. xyl*	DS (21.3%), DDS (59.7%), DTS (10.9%)	[28]
	*M. inc*	DTS (66.7%), DDS (21.3%)	[48]
	*M. jav*	DDS (31.31%), DTS (26.58%), MAT (12.25%)	[78]
*Majorana hortensis*	*C.* spp.	T-4 (41.6%), ϒ-T (13.0%), LIM (10.4%)	[37]
	*R. ren*		
	*H.* spp.		
*Mentha spicata*	*C.* spp.	CAR (58.14%)	[37]
	*R. ren*		
	*H.* spp.		
	*M. jav*	CAR (71.9%) LIM (14.3%)	[70]
*Ocimum vulgare*	*B. xyl*	CRC (35.7%), ϒ-T (23.5), σ-CYM (13.8%)	[34]
	*N. abb*	CRC (40.0%), THY (28.1%), σ-CYM (13.6%)	[68]
*Satureja hellenica*	*M. inc*	p-CYM (27.5%), CRC (23.3%)	[64]
	*M. jav.*		
*Satureja montana*	*B. xyl.*	CRC (40.0%), p-CYM (20.0%), THY (15.0%)	[35]
	*M. inc*	CRC (58.0%), p-CYM (33.0%)	[76]
*Thymus vulgaris*	*B. xyl*	No chemical analysis	[31]
	*C.* spp.	p-CYM (40.50%), THY (19.02%), CRC (14.53%)	[37]
	*R. ren*		
	*H.* spp.		
**Chemical structure**			
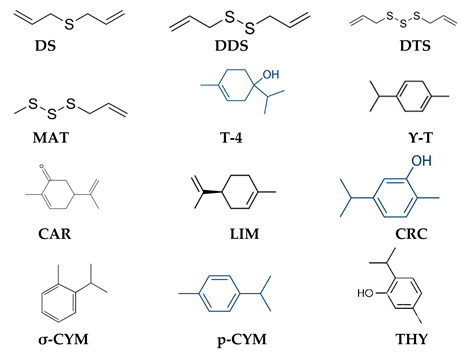			

Notes: DS = diallylsulphide, DDS = diallyl disulphide, DTS = Diallyl trisulphide, MAT = methyl allyl trisulfide, T-4 = Terpinen-4-ol, ϒ-t = ϒ-terpinene, LIM= limonene, CAR = carvone, CRC = carvacrol, THY = thymol, σ-CYM = σ-cymene, p-CYM = p-cymene.

## Data Availability

Not applicable.

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
