# Peer review of "Essential Oils as Nematicides in Plant Protection—A Review"

_plants, 2023, doi:10.3390/plants12061418_

Round 1

Reviewer 1 Report

he MS ID: plants-2280642 reviews an interesting and very important topic on the use of essential oils as an alternative method for controlling economically important parasite nematode species such as Meloidogyne incognita (Mi) and Bursaphelenchus xylophilus (Bx). Generally, the paper is well-written and presents an interesting and complex perspective on using essential oils in plant protection. I have just a few comments and suggestions for authors.

The title should be shorter and more addressed to the topic e.g. Essential oils as anti-nematode agents - A Review. Or Essential oils as nematicides used in plant protection - A Review

Page 6 line 6: Both species Mi and Bx are plant parasites, but I do not think that Bx affects food security. Bx infects only pine trees and causes economic losses rather than agricultural problems that can reflect in food security.

Table 2. In the first column, I would recommend stating as the most important information the name of nematode species as target organisms and continue in the second column with the Botanical species of EO etc.

Table 3. Latin names of plant species should be written as full botanical names because readers will only guess what you mean e.g. A. absinthium is Artemisia absinthium? I must search the internet for the species that you probably mean. The title of this table should be also changed you mention plant species in the first column and in the Title of the table you write "Names of essential oils" It is very confusing.

The English used in the manuscript is not bad, I understand what the authors probably wanted to say. Still, I would recommend sending the manuscript for professional correction, because some sentences are not clear.

Author Response

First of all, thank you for your valuable feedback. We accept all your comments. Please see below our replies. 

Rev: The title should be shorter and more addressed to the topic e.g. Essential oils as anti-nematode agents - A Review. Or Essential oils as nematicides used in plant protection - A Review

Author reply: Thank you for your comment. We agree with the referee and we changed the title as suggested: Essential oils as nematicides in plant protection – a review

Rev: Page 6 line 6: Both species Mi and Bx are plant parasites, but I do not think that Bx affects food security. Bx infects only pine trees and causes economic losses rather than agricultural problems that can reflect in food security.

Author reply: We agree with the referee that Bx only affects pine trees and it has not demonstrated impacts on food security yet. For this reason we revised the sentence accordingly to the referee suggestion from line 167 to 169.

Rev. Table 2. In the first column, I would recommend stating as the most important information the name of nematode species as target organisms and continuing in the second column with the Botanical species of EO etc.

Author reply: We agree with the referee and table 2 was revised as suggested.

Rev. Table 3. Latin names of plant species should be written as full botanical names because readers will only guess what you mean e.g. A. absinthium is Artemisia absinthium? I must search the internet for the species that you probably mean. The title of this table should be also changed you mention plant species in the first column and in the Title of the table you write "Names of essential oils" It is very confusing.

 Author reply: Thank you for your comment table 3 was revised as you suggested. The title changed in line 344.

Rev. The English used in the manuscript is not bad, I understand what the authors probably wanted to say. Still, I would recommend sending the manuscript for professional correction, because some sentences are not clear.

Author reply. A mother tongue scientific expert revised, according to your suggestion, the manuscript. 

Reviewer 2 Report

The review "A systematic review on the use of essential oils as nematicides for future application in agriculture: which one for which nematode" is much interesting and well-written but some points required more attention by the authors.

Among these points:

1. Although the idea of the review is very interesting, but I was expecting more analysis of the references if the doses used or ok  or not, the techniques used to assess the nematicidal activity. the authors own comments are highly required 

2. The authors should also correlate this activity with some of the major components in the essential oils

3. The future perspectives should be increased to highlight what s still missed to be done and the recommendations of the authors 

Author Response

Dear reviewer, first of all, thank you so much for taking the time to revise and leave your valuable and useful comments, which we accept almost all. Please find below the answer for each point raised:

Rev Number 1. Although the idea of the review is very interesting, I was expecting more analysis of the references if the doses used or ok  or not, the techniques used to assess the nematicidal activity. the authors own comments are highly required 

Author reply: Thank you for your comment. The authors of this review aim to give a tool to those who want to face for the first time the topic; the use of essential oils against nematodes, thanks to which is possible to immediately visualize which is the best essential oil to which nematode. In this view, the neophyte should find the best solution and check it in order to improve or try it for other experiments. Especially in the field, there is a very lack of information on the potential effects of essential oils on nematodes. The authors limit the personal comments to the conclusion section from line 364 to line 382.

Rev Number 2. The authors should also correlate this activity with some of the major components in the essential oils

Author reply: in the text, there are references to the activity of essential oils cited correlated to their major components that were analyzed in the various studies. For example:

-the activity of T. satureioides is correlated to the presence of thymol (line 253).

-The activity of thymol and carvacrol is also remarked in lines 328-331.

-the activity of M. spicata and T. vulgaris is correlated to the presence of components as state in lines 258-261.

However, to immediately visualize and summarise which are the main components of essential oil that determine nematicidal activity, an additional table was done. Table 4 is cited in line 367 and it is found at the end of the text. The aim of the table is to describe the main chemical components of EOs to which the highest nematicidal activity is linked, correlating to the percentages and the chemical formulas.

Rev Number 3. The future perspectives should be increased to highlight what s still missed to be done and the recommendations of the authors 

Author reply: the conclusion section has been revised as suggested by the referee. The integration has been done from line 364 to line 382.